# Mesoscale Simulations of Polymer Solution Self-Assembly: Selection of Model Parameters within an Implicit Solvent Approximation

**DOI:** 10.3390/polym13060953

**Published:** 2021-03-19

**Authors:** Juhae Park, Abelardo Ramírez-Hernández, Vikram Thapar, Su-Mi Hur

**Affiliations:** 1Department of Polymer Engineering, Graduate School, Chonnam National University, Gwangju 61186, Korea; jhp415@jnu.ac.kr; 2Alan G. MacDiarmid Energy Research Institute & School of Polymer Science and Engineering, Chonnam National University, Gwangju 61186, Korea; 3Department of Biomedical Engineering and Chemical Engineering, The University of Texas at San Antonio, San Antonio, TX 78249, USA; 4Department of Physics and Astronomy, The University of Texas at San Antonio, San Antonio, TX 78249, USA

**Keywords:** solution self-assembly, implicit solvent coarse-grained model, parameter-properties relationship

## Abstract

Coarse-grained modeling is an outcome of scientific endeavors to address the broad spectrum of time and length scales encountered in polymer systems. However, providing a faithful structural and dynamic characterization/description is challenging for several reasons, particularly in the selection of appropriate model parameters. By using a hybrid particle- and field-based approach with a generalized energy functional expressed in terms of density fields, we explore model parameter spaces over a broad range and map the relation between parameter values with experimentally measurable quantities, such as single-chain scaling exponent, chain density, and interfacial and surface tension. The obtained parameter map allows us to successfully reproduce experimentally observed polymer solution assembly over a wide range of concentrations and solvent qualities. The approach is further applied to simulate structure and shape evolution in emulsified block copolymer droplets where concentration and domain shape change continuously during the process.

## 1. Introduction

Self-assembly of polymer systems has been a topic of considerable attention in recent decades due to its relevance to many advanced nanotechnologies, such as drug delivery [1,2], medical imaging [3], nanoelectronics [4], and phononic or photonic devices [5,6,7]. Thus, extensive theoretical and numerical studies have been undertaken to understand the underlying physical principles [8,9,10,11,12]. Simulations with the atomistic resolution are not a viable option, even with the continuing advances in computing platforms, due to the large discrepancies in time and length scales between the model and the obtained self-assembled structures. Therefore, a range of coarse-grained models in which atoms are lumped together into coarse-grained segments has been developed [8,12,13,14]. One such class of models is bottom-up models that are specific to systems of interest and attempt to retain as much detail as possible about the polymers under study; these models involve conducting parameterization studies to derive effective potentials by encoding information obtained from atomistic simulations [8,9,14,15]. Because of the scale separation between monomeric repeat units on the atomistic scale and coarse-grained segments, performing such parameterization studies with sufficient accuracy remains a daunting task. One other important class is top-down models, the validity, of which at a coarser scale hinges on the universality of macromolecular systems—that is, systems with significant differences in their chemistries and microscopic interactions show similar qualitative behaviors. Physics-based key interactions represented either through particle- or field-based approaches have successfully been applied to describe how changes in enthalpic and entropic contributions affect structure formation [8,11,16].

Despite its benefits, this type of top-down approach faces challenges to accommodate recent trends in the direction to add complexities that comprise systems with complicated interactions among a large number of polymeric components of diverse types and molecular architectures along with their interactions with solvent molecules [11,17,18]. Interactions between polymers and solvent molecules may result in a wide variety of complex phenomena, e.g., swelling, deswelling, or plasticizing behaviors due to non-trivial polymer/surrounding solvent interfaces and highly sensitive self-assembled structures [19,20,21], the nature of which depends on the amount and strength of selectivity of a solvent. One example of complex systems involves the confined assembly of polymers in evaporative emulsified droplets [22,23,24,25,26]. In such systems, the inherent non-equilibrium process of continuous solvent evaporation from the soft, deformable droplet poses considerable challenges for researchers attempting to model the system and coherently represent all relevant parameters.

Even though explicit solvent top-down models allow for a more intuitive understanding of solvent interactions with polymers [27,28,29,30,31], increasing the computational overhead makes simulations computationally infeasible. Within an implicit solvent approximation, a coarse-grained model in the framework of field theory, in which energy functional is approximated in terms of the local order parameter of density fields, has been shown to qualitatively describe the self-assembly behavior in either incompressible or compressible polymer systems. The model with a generalized viral equation of local densities up to the third-order provides the flexibility to describe the qualitative features of the phase behavior of compressible mixtures. This phenomenological model has been developed in conjunction with particle-based Monte Carlo (MC) and multi-body dissipative particle dynamics (MDPD) simulation strategies [32,33,34,35,36,37,38,39,40,41].

However, within implicit solvent models, in which effective interactions between polymer and solvent are modeled without the presence of solvent particles, the selection of interaction parameters is not trivial. Previous works utilizing the virial model are limited to small sets of parameters applicable over limited ranges of solvent properties and concentration, and in some cases, justification of the choice of model parameters is lacking. As modern synthetic and fabrication methods provide a much broader palette of constituent polymer system components, selecting appropriate model parameters is of great importance.

In this report, we present our efforts to expand the capability of an implicit solvent virial model by obtaining correlations between model parameters and experimentally measurable quantities, such as single-chain scaling exponent, average chain density, and interfacial and surface tensions. The strength of the methodical model parameter selection based on the obtained parameter map is demonstrated by reproducing phase behavior in block copolymer (BCP) solutions over a wide range of polymer concentrations and solvent qualities and simulating structure and shape evolution in emulsified BCP droplets. We approximate the effect of dilution by making the model parameters dependent on the average concentration of polymer, and the solvent evaporation process is captured by changing the parameters on the fly during a simulation.

## 2. Model and Methods

### 2.1. Coarse-Grained Model with a Generalized Hamiltonian

Interactions in physics-based coarse-grained models of macromolecules comprise two major categories, bonded interactions and nonbonded interactions. The model adopted here is a coarse-grained implicit solvent model in which polymer chains are explicitly modeled with coarse-grained beads connected with harmonic springs, while nonbonded interactions are modeled in terms of density fields. In a system of *n* polymeric chains, each chain is represented by a total of *N* beads. As commonly done [42], the bonded interactions are derived from harmonic springs attached to adjacent beads in a given chain. The total harmonic potential at a given temperature, *T*, for the above system is defined as:(1)$$H_{b}=\frac{3k_{b}T}{2}\frac{N-1}{R_{e}^{2}}\overset{n}{\underset{k=1}{\sum }}\overset{N-1}{\underset{i=1}{\sum }}b_{k}^{2}\left(i\right)$$
where ***b****_k_*(*i*) is a vector connecting the *i*th and (*i* + 1)th bead in a chain, *k*. *R_e_* is the mean-squared end-to-end distance of an isolated chain with only bonded interactions present, and *k_b_* is the Boltzmann constant.

The crux of this model lies in defining nonbonded interactions. Using the virial equation of state approach, nonbonded interactions are expressed in terms of weighted number densities up to the third-order and are given by the following equation [38]:(2)$$H_{nb}=k_{b}T\int_{V}^{}\frac{d^{3}r}{R_{e}^{3}}\left(\overset{m}{\underset{\alpha =1}{\sum }}\overset{m}{\underset{\beta =1}{\sum }}\frac{v_{\alpha \beta }\rho_{1\alpha }\rho_{2\beta }}{2}+\overset{m}{\underset{\alpha =1}{\sum }}\overset{m}{\underset{\beta =1}{\sum }}\overset{m}{\underset{\gamma =1}{\sum }}\frac{\omega_{\alpha \beta \gamma }\rho_{1\alpha }\rho_{3\beta }\rho_{3\gamma }}{3}\right)$$
where integration is performed over the entire volume, *V*, of a simulation box and the summation within the integral is done over all different types of species present in a system. The total number of different types of species present in a system is labeled *m*. The number-weighted first-, second- and third-order local densities, $\rho_{1\alpha }$, $\rho_{2\alpha }$ and $\rho_{3\alpha }$, respectively, are explained in detail in the next paragraph. The subscript α in these entities refers to species type. $v_{\alpha \beta }$ and $\omega_{\alpha \beta \gamma }$ are the second- and third-order virial coefficients, respectively, which describe the interaction between species, α, β and γ, in contact with a solvent.

There are two common ways to numerically estimate local densities: grid-less-based or grid-based approaches [38,42,43]. Under both approaches, the values of local densities are inferred from bead positions. The grid-based approach involves the use of a particle-to-mesh (PM) technique, where a regular grid is introduced, and local densities are defined on each grid site [42]. In the grid-less approach, on the other hand, an off-grid continuous density is estimated by associating a density cloud with each bead [38,43]. In detail, in a grid-based approach, the simulation box is split into *M* number of regular grid sites with spacing between adjacent sites in all three directions, defined as Δ*L_G_*, and species density is estimated at these sites. A given bead at position **r** has a density contribution at one of the *M* sites, labeled the *p*th site. The contribution is given by *W*(**r** − **r**(*p*)), where **r**(*p*) is the position of the *p*th site and *W*(**r**) is the assignment function. The densities, $\rho_{1\alpha }$, $\rho_{2\alpha }$ and $\rho_{3\alpha }$, at each site are defined using the same formula and are given by:(3)$$\rho_{l\alpha }\left(p\right)=\frac{R_{e}^{3}}{N}\overset{nN}{\underset{i=1}{\sum }}W\left(r_{i}-r\left(p\right)\right)\delta_{\alpha t\left(i\right)}$$
where the summation runs over all beads, and *i* is from 1 to 3 to represent $\rho_{1\alpha }$, $\rho_{2\alpha }$ and $\rho_{3\alpha }$. *t*(*i*) denotes the species of the *i*th bead, and **r***_i_* is the position of the *i*th bead. For the assignment function, *W*(**r**), used in this work, the zeroth-order scheme is used, which means that the entire weight of a bead is assigned to the nearest grid site, regardless of its position relative to the site. This type of scheme is commonly referred to in earlier works as PM0 [42]. The prefactor chosen here is the scaling constant to ensure that density does not depend on the number of beads per chain.

In the grid-less approach, each bead is assigned a cloud density of the form $g\left(r_{i}-r\right)\delta_{\alpha t\left(i\right)}$, where *g* is the cloud function. In detail, the following different functional forms are used for $\rho_{1\alpha }$, $\rho_{2\alpha }$ and $\rho_{3\alpha }$ [38]:(4)$$\rho_{1\alpha }\left(r\right)=\frac{R_{e}^{3}}{N}\overset{nN}{\underset{i=1}{\sum }}\delta \left(r_{i}-r\right)\delta_{\alpha t\left(i\right)}$$
(5)$$\rho_{n\alpha }\left(r\right)=\frac{R_{e}^{3}}{N}\overset{nN}{\underset{i=1}{\sum }}w_{n}\left(\left|r_{i}-r\right|\right)\delta_{\alpha t\left(i\right)}$$
where *n* = 2 and 3 for second- and third-order densities, respectively. The first-order density, $\rho_{1\alpha }$, is simply a summation of delta functions; it has a nonzero value at the positions where the beads are located and a value of zero for all other locations. Introducing $\rho_{1\alpha }$ enables us to decompose the force contribution from the third-order terms (three-body interactions) into pairwise ones. The weighted second- and third-order densities ($\rho_{2\alpha }$ and $\rho_{3\alpha }$) are defined through weighting functions, *w_n_*, which are differentiable and vanish to 0 beyond a certain interaction range Δ*L_GL_*. Typically, in liquid state theory, second-order terms in the virial equation of states (Equation (2)) represent attraction, and third-order terms represent repulsion [44,45]. To represent the harsh short-range repulsions and soft long-range attractions in liquids, we choose different weighting functions, *w*_2_ and *w*_3_. Following previous studies [38], the longer-range weighting function, *w*_2_, and shorter-range weighting function, *w*_3_, are defined as:(6)$$w_{2}\left(r\right)=A\{\begin{array}{c} {\left(\Delta L_{GL}-a\right)}^{3}\;\mathrm{for}\;r\leq a\\ 2r^{3}-3\left(a+\Delta L_{GL}\right)r^{2}+\\ 6a\Delta L_{GL}r-3a\Delta L_{GL}{}^{2}+\Delta L_{GL}{}^{3}\;\mathrm{for}\;a<r<\Delta L_{GL} \end{array}$$
(7)$$w_{3}\left(r\right)=\frac{15}{2\pi \Delta L_{GL}{}^{5}}{\left(\Delta L_{GL}-r\right)}^{2}$$
where $A=\;-15/\left[2\pi \left(2a^{6}-3a^{5}\Delta L_{GL}+\;3a\Delta L_{GL}{}^{5}-2\Delta L_{GL}{}^{6}\right)\right]$ and *a* = 0.9Δ*L_GL_*.

The use of continuous and differentiable off-grid densities in a grid-less approach allows a straightforward way to estimate the forces on each bead and, as a result, to calculate thermodynamic quantities like pressure and local stress. This allows us to perform molecular dynamic simulations under any choice of ensembles, such as NVT and NPT [38,43]. However, in the grid-based technique, the use of a grid makes the computation of forces on each bead impossible. In the grid-less approach, calculating the nonbonded energy of a certain segment of beads involves explicitly computing the interactions between the segment and its neighbors, whereas in a grid-based approach, the energy of a segment is computed from the known grid-based density, which is essentially assigning a segment to its nearest site; therefore, making a grid-based approach easier to implement with a computational advantage over a grid-less approach [8,46,47].

### 2.2. Simulation Methods

Both MC and MD simulations can be performed to simulate the model described in the results section. While MC simulations are applied in conjunction with both grid-based and grid-less approaches, MD simulations are only used with grid-less approaches as the required estimation of forces is more straightforward. The implementation details for each of these methods are given below in two separate sections, MC and MD.

### 2.3. MC-G and MC-GL Simulations

MC simulations in canonical (NVT) ensembles using a grid- and grid-less-based approaches are referred to as MC-G and MC-GL simulations, respectively. For both MC-G and MC-GL simulations, to evolve a system to equilibration, we propose a trial move, which is a standard move used for equilibration of polymeric systems. Examples include bead displacement and reptation moves. The proposed trial move is accepted with probability = min(1, e^−^^Δ*H*^), where Δ*H* is the difference between the initial and final energies (Metropolis criteria). In a bead displacement move, we randomly select a bead and propose a move in all three directions by random amounts in the range of [−*dq*, *dq*]. In a reptation move, we randomly select a chain and propose a motion in which randomly chosen *s* beads are selected from either end of the chain and then reattached to the opposite end of the chain. If not explicitly stated otherwise, *dq* is set as 0.8*b*, where *b* is the mean squared bond length of an ideal chain, and the maximum value of *s* is 5. An MC cycle is then defined as a cycle, which contains a given number of these trial moves, which is dependent on the system under investigation (see Section 3). The numbers of MC cycles for equilibration of different systems are also given in Section 3. Both MC-G and MC-GL simulations are conducted using an in-house serial code written in C++.

### 2.4. MD-GL Simulations

MD simulations in the NVT ensemble combined with a grid-less approach are referred to as MD-GL simulations. For MD-GL simulations, the evolution of a system is performed by estimating forces at a given time and then updating the positions and velocities of the system using the velocity Verlet algorithm. The force contributions from bonded interactions are obtained by taking the derivative of Equation (1) with respect to the bead positions. The force contribution from nonbonded interactions is derived from the density-dependent Hamiltonian defined in Equation (2). The nonbonded force on bead *i* is obtained from taking the derivative of Equation (2) with respect to the position of the *i*th bead and can be disintegrated into the sum of pairwise forces and written as [38]:(8)$$F_{nb,i}=\frac{k_{b}TR_{e}^{3}}{N^{2}}\underset{j}{\sum }{\hat{r}}_{ij}\left[v_{t\left(i\right)t)\left(j\right)}w_{2}^{\prime }\left(\left|r_{j}-r_{i}\right|\right)+\frac{2w_{\alpha t\left(i\right)t)\left(j\right)}}{3}w_{3}^{\prime }\left(\left|r_{j}-r_{i}\right|\right)\left(\rho_{3\alpha }\left(r_{i}\right)+\rho_{3\alpha }\left(r_{j}\right)\right)\right]$$
where the summation is performed over all neighbors of bead *i* (i.e., all beads, which are within the interaction range, $\Delta L_{GL}$, of bead *i*), and ${\hat{r}}_{ij}$ is a unit vector connecting bead *i* and its neighbor, *j*.

Dissipative particle dynamics (DPDs) are combined with MD to maintain a constant temperature and preserved momentum. In this work, the dissipative and random forces present in DPD [48] vanish beyond $\Delta L_{GL}.$ For distances between particles *i* and *j*, $r_{ij}$ < $\Delta L_{GL}$, DPD forces are given by:(9)$$F^{D}\left(r_{ij},v_{ij}\right)=-\gamma \omega^{D}\left(r_{ij}\right)\left(r_{ij}\cdot v_{ij}\right){\hat{r}}_{ij}$$
(10)$$F^{R}\left(r_{ij},v_{ij}\right)=\xi \omega^{R}\left(r_{ij}\right)\theta_{ij}{\hat{r}}_{ij}$$
where $F^{D}$ and $F^{R}$ are the dissipative and random forces, respectively, and $v_{ij}$ is the vector difference between the velocity of beads *i* and *j*. According to the fluctuation-dissipation theorem [48,49], the noise coefficient, ξ, is related to the friction coefficient, γ, as $\xi^{2}=2\gamma k_{b}T$. $\theta_{ij}$ is a uniformly distributed random number in the range [−1,1]. The standard weighting functions are used for $\omega^{D}$ and $\omega^{R}$, as defined previously [48]. If not explicitly stated otherwise, the values of the MD time step and γ are used as 0.005 and 4.5. The value for the number of MD steps needed to reach equilibration in different systems is given in Section 3.2. MD-GL simulations are performed by incorporating the virial nonbonded interaction potential, *H_nb_*, into the source code of highly optimized object-oriented many-particle dynamics (HOOMD-blue), a general-purpose particle simulation toolkit optimized for execution on both graphics processing unit (GPUs) and central processing unit (CPUs) [50].

## 3. Results

### 3.1. Single-Chain Behavior in a Dilute Solution: Scaling Analysis

We examine the applicability of our coarse-grained implicit solvent model over a full range of solvent qualities, from extremely poor (including melt condition) to very good solvents. A connection between a given pair of second- and third-order virial coefficients (*v* and *w*) and solvent quality is established using simple scaling analysis. Grid-less Monte Carlo (MC-GL) simulations of an isolated homopolymer chain (type A) in a dilute solution are conducted to estimate the scaling exponent, *α*, in the power-law relation $R_{g}\propto N^{\alpha }$ where *R_g_* is the radius of gyration and *N* is the degree of polymerization. The estimated *α* value allows us to infer the solvent quality based on the well-known relation stating that, for a flexible polymer chain in dilute solution, *α* = 0.588 under good solvent conditions, *α* = 1/2 under theta (Θ) solvent or ideal chain conditions, and *α* = 1/3 under poor solvent conditions [51]. Estimation of *α* for a given set of virial coefficients (*v_aa_*, *w_aaa_*) involves obtaining average *R_g_* values for systems in which *N* ranges from 64 to 1960 and interpolating those values as shown in Figure 1b. In detail, for a given *v_aa_* and *w_aaa_*, MC-GL simulations are conducted for single-chain systems of different *N*. The initialization for each simulation is carried out by randomly placing *N* beads with bonds between adjacent beads of length *b* in a periodic box of *L_x_* = *L_y_* = *L_z_* = 10 *R_e_*. The values of *b* and *R_e_* are set as the value of bond length and mean a squared end to end distance of an ideal chain of 64 beads, respectively; those values are basic lengths units used in all the simulations reported in this work. The value of the interaction range Δ*L_GL_* is set to be 0.16*R_e_*. To measure the size of a chain after equilibration (obtained after at least 500,000 MC cycles), the value of the radius of gyration, *R_g_* averaged over 100,000 MC cycles, is calculated for each system. The scaling exponent is then estimated by fitting simulation data points at a fixed solvent quality. The scaling analysis is performed for values of *v_aa_* ranging from −4.0 to 2.0 and values of *w_aaa_* ranging from 0.005 to 0.09. 

Figure 1a shows changes in *α* for different combinations of *v_aa_* and *w_aaa_*. The obtained colormap shows the complete range of solvent quality, i.e., from a good solvent, represented by the red region, to a Θ solvent region, shown in yellow, and a poor solvent region, in blue. In the Θ solvent region, the attractive and repulsive forces among beads, which are represented by negative values of *v_aa_* and positive values of *w_aaa_*, respectively, completely compensate for each other; as a result, the ratio <*R_g_*>/*R_g, ideal_* (where *R_g, the ideal_* is the radius of gyration of an ideal chain) is approximately 1, as shown in Appendix A. A chain conformation under a set of Θ conditions is shown in Figure 1c. However, in the good solvent region, where the net interactions among polymer beads are repulsive in nature, a “swollen” structure is observed (Figure 1c). Swelling of a random coil can be captured quantitatively as, for the same *N*, Figure 1b shows a larger radius of gyration for the good solvent condition as compared to the Θ solvent condition. Conditions that produce net attractive forces are represented by the poor solvent region; such conditions cause the collapse of the coil to a globule structure with a finite density. The collapsed globule structure is shown on the far right in Figure 1c. As expected, the radius of gyration of a globule structure is smaller than that of a coil structure for the same *N* (Figure 1b). As shown in earlier studies [52,53,54], we also observed that, due to the finite size of the chains, there is no sharp transition from the coil to the globule structure. Identifying the exact boundary (tricritical Θ point) involves performing scaling studies for very-long-chain lengths [55] and is not the goal of this work. The above scaling studies were repeated using a computationally cheaper grid-based Monte Carlo (MC-G) simulation; the corresponding colormap is shown in Appendix A; as can be noticed, there are slight discrepancies in the locations of the poor theta and good solvent regions.

Along with the scaling analysis that fractionalizes the parametric space of (*v_aa_*,*w_aaa_*) to regions of different solvent qualities, further refining and narrowing down of our search of an optimal (*v_aa_*,*w_aaa_*) set can be performed by understanding the relationship of those values with other important experimentally measurable physical quantities. Configurational properties, such as the radius of gyration, are of great importance in the behavior of a polymer molecule in a solution. Thus, we begin by measuring the effect of (*v_aa_*,*w_aaa_*) on the radius of gyration using single-chain MC-GL simulations. The ratio of the relative average radius of gyration at various (*v_aa_*,*w_aaa_*) to the size of an ideal chain, <*R_g_*>/*R_g,ideal_* for a chain length of *N* = 64, is shown in Appendix A. In contrast to the sudden changes observed in the behavior of the scaling exponent, *α*, gradual changes in chain size are observed with a size ratio of approximately 1 in the Θ solvent region, >1 in the good solvent region, and <1 in the poor solvent region. Appendix A presents the degree to which a solvent is good or poor, based on (*v_aa_*,*w_aaa_*). Polymers in poorer solvents in the dark blue region in Appendix A have smaller <*R_g_*> values. Similarly, the <*R_g_*> values are larger (i.e., more swelling of polymers) in the dark red region in Appendix A, where polymers are more soluble than in the light red region.

### 3.2. Collective Properties: Surface Tension and Chain Density

We aim to expand the scope of the model from single-chain systems of dilute solutions into semi-dilute or concentrated polymer solutions and melts. Surface tension and its balance with interfacial tension, along with chain conformational entropy, play important roles in many cases, especially when the polymer solution phase separates into a polymer-concentrated and polymer-dilute phase with non-trivial geometry. Polymer melts that coexist with a vapor phase also can form non-flat free surfaces in terraced thin-films or meniscuses in confined block copolymer (BCP) films [56,57]. Recently, we reported that BCP thin-film topology at a free surface acts as a guiding template to direct the assembly of perpendicular lamellae (boundary-directed epitaxy) [56]. In a field dedicated to building complex anisotropic microparticles from BCPs under the surrounding liquid solvent medium, previous works [22,23,24] have shown that by tuning the surface interactions of the A and B blocks in AB diblock systems, one can prepare either ellipsoid or convex lens-shaped particles. Here, we explore the relationship between the model parameters and experimentally measurable tension values and chain densities. Thus, we are able to mimic the experimental conditions using the provided comprehensive tool for selecting model parameters that reflect essential physical phenomena.

The free energy of a phase-separated system has a volume and a surface contribution. The volume contribution depends on the density of polymer-concentrated phases, whereas the surface contribution depends on the surface energy/tension of the system. Therefore, the dependence of two entities, chain density (*ρ*) and surface tension (*σ*), on the strength of intermolecular interactions, which is controlled by two virial coefficients (*v_aa_*, *w_aaa_*) in a homo-polymeric solution, is explored using MD-GL simulations. The solution is prepared by placing a homopolymer film at the center of the periodic box. The initialization of a system is carried out by placing 3200 homopolymer chains confined in a thin-film of thickness 1*R_e_*, at a center of the box of thickness 10*R_e_*. Each of the homopolymer chains has *N* (=64) coarse-grained beads and is initialized with randomly positioned bonds between adjacent beads of bond length *b*. The value of the interaction range Δ*L_GL_* is set to be 0.16*R_e_*. The box dimensions in *x* and *y*-direction are 5*R_e_*. The size of the box along the *z*-direction is large enough to yield a thin-film geometry surrounded by empty regions representing implicit solvent. Periodic boundary conditions are applied in all three directions. For a given set of virial coefficients (*v_aa_*, *w_aaa_*), MD-GL simulations with a time step value Δ*t* = 0.005 are conducted. A total of 200,000 MD steps are used to equilibrate the system. The obtained configuration after equilibration is used to estimate the chain density. Along the *z*-direction, the simulation box is discretized into bins of size 0.016*R_e_*, and the number of beads in each bin is computed. The mean value of *z* over, in which the density is homogeneous (i.e., mean over the z-values where density is in between its maximum value and 80% of its maximum value), is then used as the average thickness of the equilibrated system. The total number of beads in a system divided by both *N* and average thickness gives us the value of chain density *ρ* (1/*R*_e_^3^). The estimation of *σ* is then carried out by calculating the difference between normal and tangential pressure tensor across an interface [48,58]

(11)$$\sigma =\frac{L_{z}}{2}<P_{zz}-\frac{P_{xx}+P_{yy}}{2}>$$
where *P_xx_*, *P_yy_*, *P_zz_* represent the diagonal components of the pressure tensor along the *x*-, *y*-, *z*-axes, respectively, and the brackets denote a time average. Due to the presence of two interfaces within the simulation box, a factor of 1/2 is considered. Using the equilibrated configuration, MD simulations are conducted for additional 100,000 steps, and the values of the pressure tensor are calculated every 20 steps. The time average of pressure tensor is then used to obtain *σ* in the unit of *ρkT/R_e_^2^*. The above process of estimating the surface tension *σ* and chain density *ρ* is repeated for different sets of (*v_aa_*,*w_aaa_*).

Figure 2a,b displays variations of *σ* and *ρ*, respectively, in the unit of *ρkT/R_e_^2^* and 1/*R_e_^3^* for different combinations of *v_aa_* and *w_aaa_* in the poor solvent region. The overall trends in *ρ* and *σ* are qualitatively similar to one another. However, the values of those variables can still be independently controlled in our simulations through fine-tuning of virial coefficients. For a given value of attractive strength, *v_aa_*, an increase in the strength of repulsive interactions (increase in *w_aaa_*) swells the polymer-rich domain, resulting in a decrease in *ρ*; this behavior of swelling is depicted in Figure 2c with both local density profiles and snapshots (insets of Figure 2c) of systems with different *w_aaa_* values at a fixed *v_aa_* value of −4. Figure 2c shows that the interfacial region becomes thicker and more diffuse, resulting in a decrease in surface tension. Similar behavior is observed when we weaken the strength of the attractive interaction (decrease in −*v_aa_*) when repulsive interactions are kept constant. The surface tension is reduced as the system approaches the conditions where effective attraction can no longer stabilize the phase-separated state. The location of zero surface tension is precisely where a polymer chain transits from the poor to Θ solvent regions [52] (i.e., undergoes an abrupt change in chain size from globule to coil in the case of a single-chain system (Section 3.1)). We also calculated the interfacial tension (*γ*) of immiscible homopolymer blends (homopolymers of species A and B) at various Flory-Huggins parameters between two polymers, *χ*. Using a similar MD-GL simulation procedure as that used for the estimation of *σ*, the *γ* values of immiscible homopolymer blends are calculated (Appendix A); full details, including the choice of all possible virial coefficients, are provided in Appendix A. Since *γ* values of binary homopolymer blends were predicted in earlier simulation studies using both a field-based model SCFT [59] and a particle-based model by Groot et al. [48], the magnitude of *γ* and *σ* can be compared with each other. Therefore, a subtle balance between interfacial and surface properties, which plays an important role in deciding the morphology of self-assembled structures, can be readily controlled in the simulation.

### 3.3. Phase Diagrams of Block Copolymer (BCP) Solutions

One advantage of our model is that it enables us to simulate the behavior of a polymer system in the entire range of concentrations and solvent qualities. We demonstrate this ability by completing the phase diagrams of AB diblock copolymers as a function of A block composition (*f_A_*) and polymer concentration (*ϕ*) and by comparing the results with previous works. In particular, our simulation results are compared with the results of comprehensive studies by Lodge and coworkers on the phase behavior of PS-b-PI in various solvents, including di-*n*-butyl phthalate (DBP: slightly PS selective), diethyl phthalate (DEP: more PS selective) and n-tetradecane (C14: PI selective) [60,61]. MC-G simulations are conducted under four different solvent conditions in the order of increasing selectivity; the phase diagrams in Figure 3a–d correspond to the good, theta, slightly poor, and poor solvents in relation to the B block, respectively, while the A block is solvable in all cases. In our implicit solvent model, the virial coefficients are chosen as a function of *ϕ* to reflect the effective interactions between beads. In the dilute system (*ϕ* ≈ 0), the virial coefficients of species A and B, (*v_aa,diilute_*, *w_aaa,dilute_*) and (*v_bb,dilute_*, *w_bbb,dilute_*), are chosen depending on the solvent quality based on the single-chain study (Figure 1a). Selection of virial coefficients at the melt condition (*ϕ* = 1) was done such that the vapor pressure of polymer molecules is negligible and the polymer has finite coarse-grained compressibility [8,38]. The virial coefficients in the melt condition are set such that they are equal for both A and B blocks and are labeled (*v_melt_*, *w_melt_*). The virial coefficients for the melt condition and for a dilute solution of each solvent are compiled in Table 1 and marked on the (*v*, *w*) colormap of scaling exponent shown in Figure 4. As the solvent content decreases from the dilute condition, the effective interactions between polymeric beads gradually recover back to those in the melt conditions. Thus, we select virial coefficients of *i* species under various solvent concentrations based on simple linear interpolation; virial coefficients as a function of polymer concentration *ϕ* vary along the line connecting points in the dilute and melt conditions as $v_{ii,\phi }=v_{ii,0}\left(1-\phi \right)+v_{ii,1}\phi$ and $w_{iii,\phi }=w_{iii,0}\left(1-\phi \right)+w_{iii,1}\phi$. The virial coefficients at any given *ϕ* are labeled (*v_ii,_**_ϕ_*, *w_iii,_**_ϕ_*). Figure 4 shows the trajectories of (*v*, *w*) vs. *ϕ* value. The cross-second–order virial coefficient between different polymer blocks at a given value of *ϕ*, *v_ab,_**_ϕ_*, is determined from the Flory–Huggins parameter between two species, *χ*, *v_aa,_**_ϕ_* and *v_bb,_**_ϕ_*, and is given by $v_{ab,\phi }=\chi N/\rho^{\prime }+\left(v_{aa,\phi }+v_{bb,\phi }\right)/2$. The cross third-order virial coefficients, *w_aab_*,*_ϕ_* and *w_bba_*, *_ϕ_*, are expressed as the arithmetic mean of *w_aaa,_**_ϕ_* and *w_bbb,_**_ϕ_*, and are given by $w_{aab}=\left(2w_{aaa}+w_{bbb}\right)/3,\;w_{bba}=\left(w_{aaa}+2w_{bbb}\right)/3$.

In each MC-G simulation, 8000 chains of A-B diblock copolymers of *χN* = 20 are placed in a periodic simulation box, whose dimensions vary depending on polymer concentration *ϕ*. Each chain is represented by *N* = 64 coarse-grained beads. In melt conditions, the size of the simulation box is *L_x_* = *L_y_* = *L_z_* = 4*R_e,_* which is determined to satisfy the preset averaged melt chain density *ρ′* equals to a total number of chains divided by volume of simulation box (*V = L_x_ L_y_ L_z_*). In conditions other than melt, the simulation box is varied along all three directions to reflect a given polymer concentration, *ϕ* (*ϕ* = 1 at melt condition). For a specific *ϕ*, the initialization of a system is carried out by placing AB chains in a simulation box, where each chain is initialized with randomly positioned bonds between adjacent beads of bond length, *b*. For the values of virial coefficients are set for a given concentration, MC-G simulations are performed. $2\times 10^{6}$ MC cycles are used for equilibration. For a more efficient equilibration, both single bead-displacement and reptation moves are implemented during our MC-G simulation.

Figure 3 presents the phase diagrams of each solvent condition. Observed phases of the sphere, cylinder, lamellae, bicontinuous, inverse cylinder, inverse sphere, and disordered are marked with red spheres, orange triangles, green diamonds, pink pentagons, blue inverse triangles, purple stars, and black squares, respectively. Order–disorder transition (ODT) and order–order transition (OOT) are also schematically shown (black curves). The phases are distinguished by visual inspection of the observed morphologies from MC-G simulations. One could obtain more precise phase boundaries by conducting structures, or regions in which the lamella and cylinder phases coexist are not differentiated from one another; rather, all of these are marked as bicontinuous phases in Figure 3 (a few representative snapshots of bicontinuous phases are shown in Appendix A). Thus, the broad ranges of lamella and cylinder phase coexistence observed experimentally in BCP solutions are represented with wide bicontinuous domains in our phase diagram.a free energy comparison and packing structure analysis. Perforated lamellae, gyroid structures, or regions in which the lamella and cylinder phases coexist are not differentiated from one another; rather, all of these are marked as bicontinuous phases in Figure 3 (a few representative snapshots of bicontinuous phases are shown in Appendix A). Thus, the broad ranges of lamella and cylinder phase coexistence observed experimentally in BCP solutions are represented with wide bicontinuous domains in our phase diagram. 

The phase diagram for the nonselective (neutral) good solvent for both blocks is presented in Figure 3a. In the melt state (*ϕ* = 1), BCP morphology transitions in the following sequence: disorder (black square), cylinder of A (orange triangle), lamella (green–diamond), cylinder of B (blue inverted triangle), disorder with increasing *f_A_*. Due to the slightly weak segregation strength between two blocks (*χN* = 20), sphere phases (red circle and purple star) are not observed here. Upon lowering the polymer concentration, *ϕ*, both blocks swell equally, and the incompatibility between blocks is screened out. This dilution of the effective segregation strength via the addition of a neutral solvent was clearly observed in the phase diagrams of PS-PI in neutral solvent DOP and PS-PI in DEP and DOP at high-temperature [61,62] (also, PS-PI in DEP and DOP at high-temperature). Upon dilution, ODT occurs in the concentrated region of 0.7 < *ϕ* < 0.9 for all block compositions and the phase diagram is dominated by the disordered state. The envelope of the ordered phases is symmetric at *f_A_* = 0.5. ODT presents around *ϕ* = 0.8 for *f_A_* = 0.5. Effective *χ**N* at this point following the simple dilution approximation of *χ_eff_N* = *ϕχN* matches well with the ODT values given by previous studies using the SCFT model [17].

Figure 3b–d presents phase diagrams of BCP solutions with increasing solvent selectivity. In Figure 3b, the solvent is under Θ condition for B block; thus, the solvent is slightly preferential to A block. Slightly selective solvents preferentially swell the A blocks, thus causing a lyotropic phase transition analogous to that seen when increasing the effective *f_A_* as solvent volume increases. Starting with *f_A_* = 0.3 and *ϕ* = 1 in Figure 3b, the observed phase sequence along the upward vertical direction (increasing solvent concentration) is cylinder of A, bicontinuous phase, lamellar, cylinder of B, and disordered state; that sequence matches with the one seen in the right horizontal direction (increasing *f_A_*) in the melt. Thus, the resulting ODT and OOT curves in the phase diagram are tilted to the left (smaller *f_A_*) in comparison to those under the nonselective solvent condition. Isothermal phase diagrams of PS-b-PI copolymers in DBP and DEP for different temperatures also show that, as temperature decreases, increasing the selectivity of solvents tilts the boundary between ordered phases. Self-consistent field theory (SCFT) simulation for BCPs in a selective solvent by Suo et al. [28] presented a similar phase diagram with tilted OOT when BCP is placed in a moderately selective solvent. Another important phase behavior is the enhanced stability of self-assembled structures due to solvent selectivity. Compared with the neutral solvent in Figure 3a, which readily reduces segregation strength upon addition of solvent, the slight change in solvent quality to B block significantly expands the ordered phase region and drastically lowers ODT concentration in Figure 3b.

Solvent quality to B block is reduced further such that it becomes a marginally poor solvent in Figure 3c, while A block remains solvophilic. One noticeable difference from Figure 3b is the phase reentrance that occurs for asymmetric block compositions. At *f_A_* = 0.3, as *ϕ* decreases, cylinder phase in the melt condition transits into the spherical and then the disordered state, as the effective *χ* decreases due to dilution. However, as selective solvents are added into the disordered solution, ordered cylinder phases reappear as the hydrophobicity of the B block overwhelms the dilution of segregation strength between A and B blocks. Upon further addition of solvent, the effective *f_A_* continues to increase, and cylinder, lamellae, inverse cylinder and inverse sphere phases are observed. Representative snapshots showing the sequence of phases at different *ϕ* values are shown in Appendix A (the phase reentrance location is marked in red). Similarly, along the line of *f_A_* = 0.7, the phase sequence of B cylinders, B spheres, disordered, and B spheres is observed as *ϕ* decreases (see snapshots in Appendix A). Phase reentrance from order–disorder–order transition in BCP solutions has been observed experimentally. McConnell and Gast observed lyotropic reentrant order–disorder–order–disorder phase transition in PS-*b*-PI solution with increasing decane solvent content [63]. Lodge et al. and Lai et al. [60,64] did not observe phase reentrance and concluded that the phase reentrance observed by McConnell and Gast [63] might have been an artifact of insufficiently equilibrated solutions. However, our simulation shows the possibility of the existence of the phase reentrance for limited values of asymmetricity and solvent quality. At the low concentration of polymers of *ϕ* = 0.1, over all BCP compositions, hydrophobic B cored sphere phases are stable. With a sufficient amount of solvent, the A corona block swells further, and the swollen domain eventually causes the ordered structures to disperse into micelles. In this study, we do not differentiate between packed spheres and micelles; instead, we classify both as inverse spheres. When solvent selectivity is further enhanced, the ordered structure is stabilized over most of the concentration regime, except in the case of highly asymmetric BCPs in concentrated solution, as shown in Figure 3d. Even in a system with low solvent content, self-assembled structures are mainly driven by solvent effects that exclude the strongly solvophobic B block rather than the incompatibility between A/B blocks, which is the main driving force under the melt condition.

### 3.4. Evolution of Emulsified BCP Droplets

Another interesting study on the microphase separation of BCP solutions involves the fabrication of internally ordered microparticles. As shown in Scheme 1a, BCP dissolved in a nonselective organic solvent is emulsified into droplets on the order of 10 nm–5 μm in the continuous surrounding solvent (surfactant/water) phase [23,25,26]. The organic solvent evaporates through the aqueous phase, thereby continuously shrinking the droplets and increasing the polymer concentration (see Scheme 1b). At a critical polymer concentration, the BCP starts to show microphase separation. The orientation and shape of the microphase in the deformable but confined microparticles vary according to the energy at the polymer/water interface, block-block interactions, confinement-induced entropy loss of the polymer chains, evaporation rate, type of organic solvent, etc. [23,25,26,65]. Thus, particles with diverse structures, including prolate ellipsoids, onion-like spheres, oblate ellipsoids, and others, were observed. A schematic illustration showing the production of either onion-shaped particles or ellipsoid particles due to tuning of the interfacial interactions between BCP and the surrounding solvent is shown in Scheme 1c,d. These novel self-assembled microparticles have potential applications, such as drug delivery, photonic devices, etc. [66,67]. However, compared to its potential and amount of experimental efforts, numerical studies are very limited. In a small simulation system box, deformation of the droplet (e.g., structure elongation in one direction) cannot be properly captured; however, a very large simulation box in an explicit solvent model must compensate undesirable computational load by surrounding solvent (water) whose moves are not in concern [65]. Hence, most previous simulation-based works were limited to investigating only the equilibrium morphologies of droplets using relatively inexpensive models, such as lattice-based models, which possibly underestimated the conformational entropy due to constrained bond lengths and relatively short-chain lengths [68,69,70].

Here, we have validated our implicit solvent model for simulating microphase separation in an evaporating emulsified BCP droplet. MC-G simulations are initiated by placing a symmetric AB diblock copolymer in a spherical volume surrounded by empty space, mimicking the experimental condition in which a BCP droplet exists in a surrounding solvent. We performed simulations in a cubic box of edge length *L_x_* = *L_y_* = *L_z_* = 15 *R_e_*, which is much larger than the volume of emulsified BCP droplet. The total number of symmetric AB polymer chains is 16,000, where each chain is represented by *N* = 64 beads. The presence of organic solvent inside the BCP droplet is captured by defining the variable *ϕ_P_*, which is the polymer concentration in the droplet. Simulations can start with a *ϕ_P_* value close to the concentration of ODT, as we are interested only in the last stage of evaporation, in which microphase separation and resulting shape changes into anisotropic droplets occur. Thus, the initial configuration is prepared at the semi-dilute region of *ϕ_P_* = 0.4, by placing AB chains in a spherical volume of radius, *r* with its central location as the center of the box. Each of the AB chains is initialized with randomly positioned bonds between adjacent beads of bond length, *b.* The bonds are placed in a way such that none of the beads go outside the spherical volume of radius, *r*. The radius, *r* is calculated as $r={\left(3/4\left(V_{{}_{\phi P}=0.4}\right)/\pi \right)}^{1/3}$, where $V_{{}_{\phi P}=0.4}$ is obtained as $V_{{}_{\phi P}=1}$/0.4. *ϕ_P_* = 1 is the BCP droplet volume in units of *R_e_*^3^, which is simply the total number of chains divided by the averaged chain density after organic solvent completely evaporates out from the BCP droplet of 128 (chains/*R_e_*^3^). The evaporation process, shown in Scheme 1, is captured in a simulation through time-varying virial coefficients. The virial coefficients are chosen to be on the line connecting the points *ϕ_P_* = 0 and *ϕ_P_* = 1; the virial coefficients for *ϕ_P_* = 0, labeled (*v_ii,0_*,*w_iii,0_*) for species *i*, correspond to the dilute condition in which an organic (good) solvent is abundantly present in a droplet and the coefficients for *ϕ_P_* = 1 correspond to a densely collapsed polymer under a poor surrounding solvent (labeled (*v_ii,1_*,*w_iii,1_*) for species *i*). The virial coefficient values are then linearly ramped up from their initial values to the final values at *ϕ_P_* = 1 during the course of a simulation, which resembles evaporation of organic solvent inside the droplet. During this simulation, no spatial variation of virial coefficients is introduced, assuming that as the organic solvent evaporates, it diffuses and redistributes quickly enough so that a uniform density is instantaneously achieved inside the droplet. *χN*, segregation strength between A and B at *ϕ_P_* = 1 is set to be 30. Up to the 5 × 10^6^ MC cycle, virial coefficients are ramped in a linear fashion from their values at the semi-dilute condition to those at *ϕ_P_* = 1. Once the ramping is finished, the system is further equilibrated for an additional 1 × 10^6^ MC cycles. In each MC cycle, both bead displacement and reptation moves are proposed.

Figure 5 presents our simulation results, which capture the effect of interfacial interactions between the droplet and surrounding medium on the shape and morphology of emulsified BCP droplets. The details of the corresponding experimental study by Hawker and coworkers are provided in the captions of Scheme 1 [22,23]. A system of AB diblock droplets with selective interfacial interactions with the surrounding solvent is realized in our simulation by setting different virial coefficients endpoints at *ϕ_P_* = 1 for the A and B blocks. The set of virial coefficients for the *ϕ_P_* = 0 and *ϕ_P_* = 1 conditions used for Figure 5 are marked in Appendix A. The morphologies at different stages of the evaporation process, along with the corresponding virial coefficient values marked on the (*v*, *w*) colormap of the scaling exponents, are shown in Figure 5. When A (red) has lower surface energy with the surrounding solvent than B (blue) (i.e., *γ_A/SS_* < *γ_B/SS_*, where SS stands for “surrounding solvent”), A block wets the surface exposed to surrounding solvent, and concentrically layered spherical onion structures are observed, as shown in Figure 5a. On the other hand, when the surrounding solvent is neutral for both blocks, the initial BCP domains are oriented perpendicular to the surface of the droplet, as shown in Figure 5b. However, since it is difficult to form perfect lamellar structures perpendicular to the surface in a spherical object, there are internal defects with curved A/B interfaces (see the morphologies labeled (2) and (3) in Figure 5b). The defective structure is gradually annihilated, and elongation in the axial direction starts to occur. Thus, for a certain concentration range, mixtures of radial layers and axially stacked lamellae (see the morphologies labeled (4) in Figure 5b) are observed; similar structures were experimentally observed between the phase transition from onion-like spherical nanoparticles to ellipsoidal stacked lamellae. As the concentration inside the droplet increases, the effective *χN* between A and B recovers back to the value in the *ϕ_P_* = 1 condition. Enhanced tension at the A/B interface due to stronger segregation strength drives the particles into a prolate ellipsoid structure with a flat A/B interface that is axially stacked.

## 4. Conclusions

Using a soft coarse-grained model combining bead-based chain representation and density field-based energy functional representation, we have proposed a simulation framework that allows for optimal model parameter selection by exploring the relationship between the parameters and important experimentally measurable quantities. In particular, the relationship between the model parameters in our implicit solvent model and solvent quality, surface (interfacial) tension, and structural (radius of gyration) and bulk properties (density) is successfully obtained. The model’s flexibility under different simulation methods was shown by using both MC and MD formalisms in grid-based and grid-less representations. Our investigation demonstrates that the suggested approach captures BCP phase behavior in solutions over a broad range of experimental conditions. Furthermore, simulations updating model parameters on the fly mimicking the self-assembly process and morphological changes in evaporating emulsified BCP droplets have shown good agreement with experimental observations. The parameter selection approach applied to an implicit solvent soft coarse-grained model may, therefore, provide a simple, efficient route to understand the collective motion of chains at large spatial and temporal scales, which then determines the kinetic and thermodynamic behavior of polymeric systems in both solutions and melts. Although we focused only on applying the model to flexible block copolymer systems, our parameter selection approach is versatile and can easily be extended to address different systems, including systems with semiflexible chains or liquid crystalline materials and systems with different chains architectures, such as bottlebrush systems. The developed model also has the potential to be extended to simulations of charged systems via the incorporation of electrostatic potential in our nonbonded Hamiltonian.

## Data Availability

The datasets generated during and/or analyzed during the current study are available from the corresponding author on reasonable request.

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
