# Peer review of "Mesoscale Simulations of Polymer Solution Self-Assembly: Selection of Model Parameters within an Implicit Solvent Approximation"

_polymers, 2021, doi:10.3390/polym13060953_

Round 1

Reviewer 1 Report

The presented study reports on the Mesoscale Simulations of Polymer Solution Self-Assembly: Selection of Model Parameters within an Implicit Solvent Approximation. And a scaling effect and model has been demonstrated by the simulations. And a proper simulation result has been presented to support the claim of this study. After carefully reading it, and I think I would like to suggest to consider the following comments to improve this study.

1. In the “Introduction” section, the following sentences of “Interactions between polymers and solvent molecules may result in a wide variety of complex phenomena, e.g., non-trivial polymer/surrounding solvent interfaces and highly sensitive self-assembled structures, the nature of which depend on the amount and strength of selectivity of a solvent.” This claim is right, but it is not professional in practical application.

Generally, we can not use the polymer/solvent interfaces and self-assembled structures to support the complex phenomena. As in the experimental measurement, these two parameters can not be observed, and they are always measured and they are analytical results. Generally, the swelling behavior, color and transparency are used to present the interaction between polymers and solvent molecules. While polymer/solvent interfaces and self-assembled structures are micro-sized experimental results, and it is difficult to identify them in practical experiments.

I suggest to revise the sentences as “Interactions between polymers and solvent molecules may result in a wide variety of complex phenomena, e.g., swelling, deswelling, or plasticizing behaviors due to non-trivial polymer/surrounding solvent interfaces and highly sensitive self-assembled structures, the nature of which depend on the amount and strength of selectivity of a solvent.” And the following references can be used to support it, i.e., (1) Menge H; Hotopf S; Ponitzsch S; Richter S; Arndt KF; Schneider H; Heuert U. Investigation on the swelling behaviour in poly(dimethylsiloxane) rubber networks using nmr and compression measurements. POLYMER. 1999, 40, 5303-5313. (2) Haibao Lu and Shanyi Du. A phenomenological thermodynamic model for the chemo-responsive shape memory effect in polymers based on Flory-Huggins solution theory. Polymer Chemistry. 2014, 5(4), 1155-1162.  (3) Haibao Lu, Yanju Liu, Jinsong Leng and Shanyi Du. Qualitative Separation of the Physical Swelling Effect on the Recovery Behavior of Shape Memory Polymer. European Polymer Journal. 2010, 46(9): 1908-1914.

2. Please present the radius of gyrations of single chain in Figure 1c, at the given Na. It is helpful to catch up with what the type of the radius of gyrations.

3. Figure 5 is not so clear to catch, please promote it with a high quality. Especially for the images, and 2D image can be used to better illustrate it.

4. The “4. Model and Methods” section is suggested to move into the section of “2.1. Model Summary”.

In all, an interesting and useful study, I would like to recommend it after the revision.

Reviewer 2 Report

In the manuscript, Park et.al utilized a hybrid solvent free model to study polymer solution; various phenomena are investigated, including the single chain behavior, surface tension, chain density and phase diagrams of block copolymers. The method is clear described. And the paper is well-written.

I  have two questions:

  • What the dynamic properties will look like with the solvent-free method. For example, what the polymer dynamics would like in the dilute solution and (melt if possible)? Whether the results in simulation are consistent with the Zimm model and Rouse model? I think it will be great if the authors can test these. Additionally, these dynamic properties are related to the evolution of the morphology change of the copolymer.
  • What the detailed structure properties will look like? Instead of see the morphology figure for a qualitative check. Is it possible to have a quantitative check (such as static structure factor ) and compare it to the experiment?
